# Fabrication of Active Z-Scheme Sr_2_MgSi_2_O_7_: Eu^2+^, Dy^3+^/COF Photocatalyst for Round-the-Clock Efficient Removal of Total Cr

**DOI:** 10.3390/molecules29184327

**Published:** 2024-09-12

**Authors:** Meng Xu, Junshu Wu, Mupeng Zheng, Jinshu Wang

**Affiliations:** Key Laboratory of Advanced Functional Materials, Ministry of Education, College of Materials Science & Engineering, Beijing University of Technology, Beijing 100124, China; xum11062020@163.com (M.X.); junshuwu@bjut.edu.cn (J.W.)

**Keywords:** photocatalyst, Sr_2_MgSi_2_O_7_, COF, Cr(VI)

## Abstract

Photoreduction is recognized as a desirable treatment method for hexavalent chromium (Cr(VI)). However, it has been limited by the intermittent solar flux and limited light absorption. In this work, a novel Z-scheme photocatalyst combining a covalent organic framework (COF) with Eu^2+^, Dy^3+^ co-doped Sr_2_MgSi_2_O_7_ (Sr_2_MgSi_2_O_7_:Eu^2+^, Dy^3+^) is synthesized, which shows the high spectral conversion efficiency and works efficiently in both light irradiation and dark for Cr(VI) reduction. Sr_2_MgSi_2_O_7_:Eu^2+^, Dy^3+^ serves as both an electron transfer station and active sites for COF molecule activation, thus resulting in 100% photoreduction of Cr(VI) (50 mL, 10 mg/L) with high light stability and over 1 h dark activity. Moreover, the XPS and FT-IR analyses reveal the existence of functional groups (Si-OH on Sr_2_MgSi_2_O_7_:Eu^2+^, Dy^3+^, and -NH- on COF_TP-TTA_) on the composited catalyst as active sites to adsorb the resultant Cr(III) species, demonstrating a synergistic effect for total Cr removal. This work provides an alternative method for the design of a round-the-clock photocatalyst for Cr(VI) reduction, allowing a versatile solid surface activation for establishing a more energy efficient and robust photocatalysis process for Cr pollution cleaning.

## 1. Introduction

Since the advent of the industrial revolution, the excessive emission of anthropogenic Cr(VI) from metallurgy, tanning, and other industrial processes has caused serious environmental concerns, including global soil pollution and underlying water resource crises [1]. Cr(VI) is a known carcinogen and mutagen that is highly toxic, whereas trivalent chromium (Cr(III)) is significantly less toxic and much less mobile in the environment [2,3]. Photoreduction of Cr(VI) into Cr(III), powered by inexhaustible but intermittent solar energy, provides a feasible pathway for the construction of “lucid waters and lush mountains”. However, the thermodynamic oxidation of Cr(III) typically occurs in natural environments, and the re-oxidation of Cr(III) species is gradually causing secondary pollution of soil and groundwater [4,5,6]. Currently, the investigation of total Cr (Cr(T), including Cr(VI) and Cr(III)) removal through the integration of photocatalysis and adsorption, is still in its infancy. The Cr(T) removal rates of many recently developed catalysts are inadequate for practical demands, and the produced Cr(III) remains in aqueous solutions [7]. To enable complete Cr(T) removal, a highly efficient photocatalyst is essential, taking into account factors such as photon capture, charge carrier separation and transfer efficiency, proton-coupled electron transfer kinetics, and adsorption kinetics on the surface of the material [8,9]. In recent decades, the number of suitable candidates for exploring photocatalytic Cr(VI) reduction has witnessed a huge surge. However, the use of a single material such as metal oxides [10], carbon nitride [11], metal–organic frameworks [12], etc., remains a challenge in meeting all these requirements [13]. In addition, the amount of time that any location on Earth can receive sunlight every day is limited, which results in the photocatalytic reactions only taking place during daylight hours and the utilization efficiency of photocatalysts being severely reduced [14]. Thus, there is still considerable scope for the development of a round-the-clock photocatalyst for the highly efficient removal of Cr(T) in the future.

Long-afterglow catalysis, also known as persistent or long-lasting catalysis, offers a conceptual solution to the excessive dependence of traditional photocatalysis on constant light irradiation [15,16]. The employment of long-afterglow materials as a photocatalyst enables the photoinduced charges to be captured by the traps during illumination. Following the termination of irradiation, the trapped electrons then flee from the traps, exhibiting long-afterglow luminescence and even enabling photo reduction around the clock [17,18]. For this purpose, several long-afterglow materials have been investigated as a kind of special phosphor and photocatalyst possessing a unique “charge storage pool” effect. More recently, Sr_2_MgSi_2_O_7_: Eu^2+^, Dy^3+^ (SMSO_ED_) has been widely used in the field of luminescent materials, which can be used to assist photocatalytic reactions [19,20]. This phenomenon may be attributed to the existence of Eu^2+^ and Dy^3+^, in which the Eu^2+^ acts as the luminescent center and Dy^3+^ is regarded as the auxiliary activator and electron trap, giving SMSO_ED_ long-afterglow luminescence properties [21]. Furthermore, silicate long-afterglow materials possess several advantageous properties, including a narrow bandgap (≈2.7 eV), excellent chemical stability, and low cost, which are thus suitable for use as photocatalysts [22]. However, current persistent photocatalysts, exemplified by SMSO_ED_, are confronted with challenges such as easy recombination of photo-generated carriers in single-component photocatalysts, low response to visible light, and short charge storage duration. To overcome these shortcomings, it is best to combine SMSO_ED_ with photocatalysts that exhibit photoactivity in the wavelength region of the matching bathochromic side. This is beneficial for maximizing photon energy transfer from SMSO_ED_ to photocatalyst, thereby improving both photocatalytic performance and energy utilization efficiency [23].

Currently, there are reports of SMSO_ED_-based round-the-clock photocatalysts for wastewater treatment, such as Sr_2_MgSi_2_O_7_: Eu^2+^, Dy^3+^/Ag_3_PO_4_, Sr_2_MgSi_2_O_7_:Eu^2+^, Dy^3+^/g-C_3_N_4_, and Sr_2_MgSi_2_O_7_: Eu^2+^, Dy^3+^/ZnO [24,25,26]. In these studies, photocatalysts like Ag_3_PO_4_ and g-C_3_N_4_ exhibit inadequate responsiveness to visible light and fail to adequately absorb the light emitted by long-afterglow materials. Covalent organic frameworks (COFs) represent an emerging class of crystalline and porous polymer semiconductors with the advantages of thermal and chemical stability, large specific surface area, an appropriate Cr(VI) reduction band gap, and a wide light-absorption region [27,28,29]. The COF_TP-TTA_, derived from TP (1,3,5-triformylphloroglucinol) and TTA (4,4′,4″-(1,3,5-triazine-2,4,6-triyl) trianiline), is a typical triazine structure COF and exhibits a high degree of light trapping ability under visible light. The presence of D-A (donor–acceptor) units could expand the delocalization range of electrons thereby enhancing the separation efficiency of electron-hole pairs [30,31]. In addition, the abundant functional groups (amino or hydroxyl) and rich pore structure endow COF_TP-TTA_ with more adsorption and activation sites for Cr(III) species. Furthermore, the Z-scheme photocatalytic system, constituted by the combination of SMSO_ED_ and COF_TP-TTA_, is capable of further inhibiting the recombination of the photo-generated carriers. Simultaneously, it exhibits a more negative CB potential and a positive VB potential, allowing the composite material to display superior redox ability. Concurrently, the light absorption range (200–600 nm) of COF_TP-TTA_ significantly overlaps with the emission range of SMSO_ED_, suggesting that the long afterglow emitted by SMSO_ED_ can be effectively absorbed by SMSO_ED_/COF_TP-TTA_ photocatalysts.

Here, a novel round-the-clock photocatalyst Sr_2_MgSi_2_O_7_: Eu^2+^, Dy^3+^/COF_TP-TTA_ (SMSO_ED_/COF_TP-TTA_) is synthesized by using natural vermiculite as silicon source to remove Cr(T) by reducing Cr(VI) and synergistically adsorbing Cr(III) during the day and night. A comprehensive study of the morphology, crystal structure, surface chemical state, and band structure reveals that integrating SMSO_ED_ with COF_TP-TTA_ can expand the optical response region and improve quantum efficiency, thereby enhancing the reactivity of the photocatalyst. Furthermore, the Cr(VI) photoreduction efficiency reaches 100% after 30 min of self-photocatalytic reaction in the absence of light, thus breaking the limitation of sunlight irradiation in photocatalytic technology. Subsequently, the products of the Cr(III) species from the photoreduction process are absorbed by the groups on the surface of the catalyst, achieving 100% removal efficiency of Cr(T). This work offers an effective strategy for the design of composited photocatalyst to purify wastewater containing Cr without sufficient sunlight. A Z-scheme charge transfer mechanism is also proposed to elucidate the enhanced photocatalytic performance.

## 2. Results and Discussion

### 2.1. Structure and Morphology Analysis

The crystal structure of the obtained photocatalysts was determined by XRD patterns, as shown in Figure 1a. The pristine SMSO_ED_ exhibits characteristic peaks centered at 28.1°, 30.3°, and 43.2°, which index with the (201), (211), and (212) lattice planes of Sr_2_MgSi_2_O_7_ (PDF#75-1736). With regard to COF_TP-TTA_, it possesses a broad peak at 25–27°, which belongs to the (011) plane of COF_TP-TTA_ and indicates Π-Π stacking of the two-dimensional layers [32]. Furthermore, all characteristic peaks can be assigned to both SMSO_ED_ and COF_TP-TTA_ in the patterns of the SMSO_ED_/COF_TP-TTA_ composites. No changes in typical diffraction peaks are observed in the XRD patterns of SMSO_ED_ and the obtained SMSO_ED_/COF_TP-TTA_ samples. This may be caused by the substitution of a portion of Sr^2+^ with Eu^2+^ and Dy^3+^ into the lattice because of the similar ionic radii of Sr^2+^ (1.18 Å), Eu^2+^ (1.17 Å) and Dy^3+^ (0.912 Å) and the low doping level of Eu^2+^ and Dy^3+^. The peaks observed in the SMSO_ED_/COF_TP-TTA_-5 sample are assigned to tetragonal SMSO_ED_ phase, as no signals attributable to COF_TP-TTA_ signals are identified. This is likely due to the low loading content of COF_TP-TTA_ in the sample. As the COF_TP-TTA_ mass ratio rises, the emergence of broad peaks of COF_TP-TTA_ is ascertained in SMSO_ED_/COF_TP-TTA_-10 and SMSO_ED_/COF_TP-TTA_-20, reflecting the effective incorporation of SMSO_ED_ and COF_TP-TTA_ via an in situ solvothermal synthesis. Furthermore, the chemical structures and functional groups of the prepared samples are analyzed in greater detail by FT-IR spectra in Figure 1b. The absorption observed at 3430 cm^−1^ arises from the symmetric stretching vibrations of hydrated hydroxyl. In the FT-IR spectrum of SMSO_ED_, the absorption peaks at 1006 cm^−1^ and 624 cm^−1^ are attributed to the Si-O-Si stretching vibration, while the peaks caused by the O-Si-O stretching vibration are located at 966 cm^−1^, 922 cm^−1^, and 836 cm^−1^ [33]. Furthermore, the bending vibrations of the Si-OH bond are observed at 563 cm^−1^, while the peak at 473 cm^−1^ is brought about by the stretching vibration of the Mg-O bond. For COF_TP-TTA_, the peaks observed at 1623 cm^−1^ and 1453 cm^−1^ can be assigned to the vibrations of the keto C=O, C=C and the aromatic ring, respectively, together with the stretching and vibrational modes of the triazine ring at 1378 cm^−1^ and 1504 cm^−1^. This evidence suggests the formation of β-ketoamine bonding in COF_TP-TTA_ [34]. Peaks attributed to C=N and C-N are observed at 1573 and 1258 cm^−1^, respectively. The FT-IR spectrum of SMSO_ED_/COF_TP-TTA_-10 displays common peaks of SMSO_ED_ and COF_TP-TTA_, and the response peak exhibits a slight increase with the increase in the COF_TP-TTA_ proportion. In addition, the absence of new functional groups suggests that the interaction between COF_TP-TTA_ and SMSO_ED_ in the composite occurs through interfaces rather than chemical bonds, which is consistent with the XRD patterns [24].

According to the SEM and TEM results, SMSO_ED_ presents an irregular block-like morphology with a smooth surface (Figure 2a,d). The original COF_TP-TTA_, by contrast, is a stacked “yarn”-like architecture (Figure 2b,e). Figure 2c,f illustrate the morphologies of the SMSO_ED_/COF_TP-TTA_-10 composite. A substantial number of COF_TP-TTA_ nanowires grow in a dense configuration, exhibiting a reduced propensity for adhesion and agglomeration. In addition, the incorporation of COF_TP-TTA_ into SMSO_ED_ results in the formation of a dense heterostructure with intense interfacial interaction, characterized by the tight adherence of numerous COF_TP-TTA_ to the surface of SMSO_ED_. The effective combination of SMSO_ED_ and COF_TP-TTA_ is exhibited in the HRTEM image, in which the lattice spacing of 0.293 nm is identified as corresponding to (211) of SMSO_ED_, while COF_TP-TTA_ shows an amorphous structure (Figure 2g). The elemental mapping images of the SMSO_ED_/COF_TP-TTA_-10 show a uniform distribution of N, O, Si, Mg, Sr, Eu, and Dy elements across the entire surface of the samples, thereby confirming the successful generation of the SMSO_ED_/COF_TP-TTA_ composite (Figure 2h).

### 2.2. Surface Chemical State Analysis

The respective XPS fine spectra of SMSO_ED_ and SMSO_ED_/COF_TP-TTA_-10 are shown in Figure 3a–f. From the XPS full spectrum, the presence of Si, Mg, Sr, O, C, and N elements can be observed. Furthermore, the peaks of Eu and Dy elements do not appear due to their low content (Appendix A). Figure 3a illustrates the high-resolution Si 2p spectra of SMSO_ED_ and SMSO_ED_/COF_TP-TTA_-10. The peaks observed at 102.1 eV and 102.8 eV are assigned to the Si-O chemical bond. The binding energy of approximately 1034.6 eV is associated with the Mg^2+^ in the SMSO_ED_. The high-resolution O 1s of SMSO_ED_ can be deconvolved into two peaks: 531.5 eV (Si-O bond) and 532.6 eV (-OH bond). The new peak at 533.9 eV is attributed to C=O, which is indicative of the integration of COF_TP-TTA_ on the SMSO_ED_ surface. The N 1s spectra of SMSO_ED_/COF_TP-TTA_-10 show the location of the C-N/N-H (399.5 eV) and C=N (398.1 eV) peaks, indicating that the surface of the composite material provides abundant functional groups for Cr species adsorption (Appendix A). The Sr 3d spectrum of these two catalysts contains two symbolic peaks (133.8 eV and 135.3 eV) associated with Sr 3d_5/2_ and Sr 3d_3/2_ spin orbits [35]. In Figure 3e, the binding energies at 1123.8 eV, 1134.6 eV, 1163.0 eV, and 1172.1 eV pertain to 3d_5/2_ and 3d_5/2_ orbitals of Eu^2+^ and Eu^3+^, respectively. The Dy4d spectrum of SMSO_ED_ and SMSO_ED_/COF_TP-TTA_-10 exhibits two peaks at 148.6 and 153.8 eV, which correspond to the Dy 4d_5/2_ and 4d_3/2_ orbitals [36]. Compared with SMSO_ED_, a positive displacement is observed in the majority of high-resolution spectra of the SMSO_ED_/COF_TP-TTA_-10 composite, while compared with COF_TP-TTA_, the N1s peaks display a negative shift (Appendix A). These results indicate that the intense interfacial interaction between SMSO_ED_ and COF_TP-TTA_ could lead to the formation of an internal electric field (IEF), which in turn promotes directional charge drift from SMSO_ED_ to COF_TP-TTA_ in SMSO_ED_/COF_TP-TTA_-10 until their E_f_ values reach equilibrium [37].

### 2.3. Energy Band Structure and Photoelectrochemical Analysis

It is widely acknowledged that energy band structures are essential for a comprehensive investigation of interface charge transfer processes. The UV-vis diffuse reflectivity test is performed to investigate the optical properties of SMSO_ED_, COF_TP-TTA_, and SMSO_ED_/COF_TP-TTA_ composites with different mass ratios (Figure 4a). It is evident that SMSO_ED_ exhibits an absorption edge at 475 nm. After the introduction of COF_TP-TTA_, the absorption edges are redshifted to 558–595 nm. Furthermore, the SMSO_ED_/COF_TP-TTA_-10 composite exhibits a more pronounced redshift and higher absorption value in the visible region than the other composites. The enhanced optical absorption potentials enable SMSO_ED_/COF_TP-TTA_-10 to use a broader spectrum of light for the generation of more reactive substances, thereby accelerating the photocatalytic process. Based on Tauc’s plot, the band gap energy (E_g_) can be obtained by fitting the optical jump at the absorption edge [38]. As illustrated in Figure 4b,c, the forbidden bandgaps of SMSO_ED_ and COF_TP-TTA_ are 2.74 eV and 2.32 eV, respectively. The Fermi energy level of these two semiconductors can be evaluated by the Mott–Schottky diagram. In addition, the flat band potentials of SMSO_ED_ and COF_TP-TTA_ are estimated to be −0.60 V and −0.67 V vs. saturated calomel electrode (SCE), which can be converted into −0.36 eV and −0.43 eV vs. NHE (pH = 7), respectively (Figure 4d,e). Commonly, the conduction band potentials (ECB) of n-type semiconductors are 0.2 V more negative than their flat band potentials [39]. Consequently, their ECB is calculated to be −0.56 and −0.63 eV, respectively. By employing the equation E_VB_ = E_CB_ + E_g_, the VB potentials of SMSO_ED_ and COF_TP-TTA_ can be calculated to be 2.18 and 1.69 eV, respectively. These data reveal the energy band structure of SMSO_ED_ and COF_TP-TTA_. As a result, as illustrated in Figure 4f, the energy bands of SMSO_ED_ and COF_TP-TTA_ are arranged in a cross pattern.

To further investigate the migration and recombination rates of photogenerated carriers of the prepared catalysts, the PL spectra are presented in Figure 5a. The pure COF_TP-TTA_ exhibits robust luminescence in a broad range (550–700 nm), which indicates the generation of photoinduced electron–hole pairs and the rapid recombination process. The broadband emission of SMSO_ED_ centered at 467 nm, can be attributed to the typical 4f^6^5d^1^-4f^7^ transition of Eu^2+^ [40]. It is noteworthy that the composite material retains the phosphorescence properties of SMSO_ED_. The phosphorescence with a wavelength of 467 nm generated by SMSO_ED_ can be employed for photocatalytic reactions on COF_TP-TTA_, given that COF_TP-TTA_ is capable of absorbing visible light up to 600 nm. For SMSO_ED_/COF_TP-TTA_-10, the PL intensities decreased obviously, indicating that the heterostructure of SMSO_ED_ and COF_TP-TTA_ effectively inhibits the recombination of photocarriers. Figure 5b presents the decay curves of fluorescence intensities of the single catalysts and the SMSO_ED_/COF_TP-TTA_-10 composite after removing light source. It can be clearly observed that each curve consists of a rapid decay followed by a slow decay with a long decay tail [41]. The SMSO_ED_ and SMSO_ED_/COF_TP-TTA_-10 demonstrate a rapid decay of the emission intensity at the beginning, followed by a slow decay in persistent intensity, indicating a long-lasting decay process. Despite the reduction in both the intensity and duration of the afterglow resulting from the addition of COF_TP-TTA_, the afterglow performance can still persist for up to an hour. Figure 5c shows the actual emission of SMSO_ED_ after simulated sunlight irradiation. The original sample is light yellow when exposed to sunlight. After the light is turned off, the irradiated SMSO_ED_ emits strong blue light, and the emission intensity gradually decreases with time, which is consistent with the results of the decay curve spectra. Combined with the above-mentioned light absorption characteristics of the long-afterglow phosphor SMSO_ED_, it can be used as an effective constant internal luminescent material in photocatalytic composite materials. The separation and transport rates of the photogenerated electrons and holes can be investigated by the photocurrent and EIS tests (Figure 5d,e). The lower the charge transfer impedance, the smaller the radius shown in the Nyquist plot. SMSO_ED_ has the largest radius and the lowest charge transfer efficiency, while COF_TP-TTA_ has a higher charge transfer efficiency. This indicates that COF_TP-TTA_ has better charge transfer capability, while COF_TP-TTA_ only accounts for 10% of the total weight of the SMSO_ED_/COF_TP-TTA_-10 composite, resulting in a slower charge transfer rate. The photocurrent intensity of COF_TP-TTA_ and SMSO_ED_/COF_TP-TTA_-10 is obviously higher than that of pure SMSO_ED_. These results suggest that the introduction of the pure COF_TP-TTA_ effectively accelerates the charge transfer and separation. In addition, the SMSO_ED_/COF_TP-TTA_-10 sample not only shows long-afterglow photocatalytic activity but also exhibits better synergistic effects in Cr(T) removal, as described below.

### 2.4. Cr(T) Removal Performance Analysis

The performance of the prepared long-afterglow catalysts for Cr(VI) photoreduction and Cr(III) adsorption was examined under simulated sunlight irradiation and in the dark without any photosensitizer or sacrificial agent. The relevant results are shown in Figure 6. Before the photoreduction, all of the samples were mixed with a certain concentration of Cr(VI) and stirred continuously for 30 min in the dark to reach adsorption–desorption equilibrium. Figure 6a,b show the performance of the photocatalysts with the varied proportion of SMSO_ED_ and COF_TP-TTA_ in the SMSO_ED_/COF_TP-TTA_ composites. As anticipated, pristine SMSO_ED_ has either weak adsorptive or reductive ability towards Cr(VI), so the concentration of Cr(VI) remains almost unchanged, and the Cr(T) removal efficiency is only 12.3%. Pure COF_TP-TTA_ can effectively adsorb Cr(VI), which is due to the positive surface potential of COF_TP-TTA_ at the initial pH = 4. Cr(VI) exists in the form of anions (HCrO_4_^−^, pH = 4), and thus the adsorption is dominated by electrostatic attraction (Appendix A). Under light conditions, COF_TP-TTA_ exhibits an excellent reduction performance towards Cr(VI) with 96.7% removal efficiency during 30 min, but the reduction process only occurs in the presence of an external light source. Within 30 min after removing the light source, the concentration of Cr(VI) increases due to the partial desorption of Cr(VI) adsorbed on the surface of COF_TP-TTA_, resulting in a stable removal efficiency of 90.9% after reaching the desorption equilibrium. In comparison, the binary materials (SMSO_ED_/COF_TP-TTA_) show more efficient removal activity in light and dark conditions. Since the surface of the composites are also positively charged at pH = 4, these composites exhibit adsorption performance for Cr(VI) to diverse degrees subsequent to 30 min of dark adsorption (Appendix A). After 30 min of photocatalysis, 55% (SMSO_ED_/COF_TP-TTA_-5), 92.8% (SMSO_ED_/COF_TP-TTA_-10), and 93.8% (SMSO_ED_/COF_TP-TTA_-20) of Cr(VI) are reduced, indicating that the heterojunction exhibits enhanced photocatalytic performance. The quasi-first-order kinetic model is used to analyze the linear transformation of the data, and the K (K is the kinetic constant) under light irradiation are 0.019, 0.067, 0.066 min^−1^ when the COF_TP-TTA_ addition amounts are 5%, 10%, and 20%, respectively (Figure 6b and Appendix A). In order to investigate the role of long-afterglow material SMSO_ED_ in the prepared photocatalysts, the effect of the continued reduction of Cr(VI) after the cessation of illumination is also shown in Figure 6a,b. As the mass ratio of COF_TP-TTA_ increases, the dark catalysis activity first increases and then decreases. The optimal mass ratio of COF_TP-TTA_ to SMSO_ED_ is 10%. When COF_TP-TTA_ accounts for a small proportion, the amount of generated carriers decreases and the COF_TP-TTA_ cannot fully accept energy from SMSO_ED_. When the proportion of COF_TP-TTA_ is too large, SMSO_ED_ cannot provide sufficient energy for effective photocatalysis in the dark. Therefore, the maximum value of K is 0.0034 min^−1^ for the optimal sample (SMSO_ED_/COF_TP-TTA_-10) under the dark conditions in this experiment (Appendix A). The removal efficiencies of Cr(VI) and Cr(T) can be achieved to 100% and 97.16%, respectively, within 60 min (Figure 6b). This further demonstrates that SMSO_ED_/COF_TP-TTA_-10 has the best photoreduction and adsorption performance among the series of samples. In order to elucidate the advantages of this work in Cr(T) removal compared to previously reported photocatalysts, the comparison of this work with similar photocatalytic systems is shown in Appendix A.

### 2.5. Effects of pH Value on Cr(T) Removal

The Cr(VI) photoreduction rate is usually influenced by the initial pH of the solution [42]. The decrease in pH will greatly facilitate the photocatalytic reduction of Cr(VI). As shown in Figure 6c,d, the effectiveness of SMSO_ED_/COF_TP-TTA_-10 in eliminating Cr(VI) reached 93.6% at a pH of 2 after 30 min of light exposure, but it progressively declined as the pH increased. This phenomenon can be explained by the forms of Cr(VI) present at different pH solutions (Appendix A). The species of Cr(VI) are expressed as H_2_Cr_2_O_7_ in a low pH, and the photocatalytic reaction of Cr(VI) occurs in the following way [43,44]:$$H_{2}{\mathrm{Cr}}_{2}O_{7}+{6H}^{+}+{3e}^{-}\to {\mathrm{Cr}}^{3+}+{4H}_{2}O$$

In alkaline solutions, the main species of Cr(VI) are CrO_4_^2−^, and the reaction proceeds as:$${{\mathrm{CrO}}_{4}}^{2-}+{4H}_{2}O+{3e}^{-}\to {\mathrm{Cr}(\mathrm{OH})}_{3}+{5\mathrm{OH}}^{-}$$

The lower pH is favorable for Cr(VI) reduction into Cr(III). Conversely, the generation of Cr (OH)_3_ precipitate is more readily achieved at a higher pH, with the activity sites becoming attached with the precipitate, thereby reducing the photoreduction activity [45]. Although neutral and alkaline conditions are more conducive to the adsorption of Cr(III), the Cr(T) removal efficiencies remain relatively low at 45.1, 12.2, and 6.2% at pH = 6, 8, and 10, respectively. This is due to the fact that the pH conditions are not conducive to the photochemical reduction reaction, and the negative surface charge of SMSO_ED_/COF_TP-TTA_-10 is also adverse for the adsorption of Cr(VI). They result in a decrease in the Cr(T) removal efficiency. At pH = 2 and pH = 4, the rate constants remain at a relatively high level, 0.067 and 0.071 min^−1^, respectively (Appendix A). However, the Cr(T) removal efficiencies are determined to be 89.4% and 97.16%, respectively, due to the competition between excess H^+^ and cationic Cr(III) in the solution with low pH value. Therefore, the removal efficiency of Cr(T) in the one-step system is dominated by Cr(VI) reduction (pH > 4) or Cr(III) adsorption (pH < 4). In addition, the pH value of actual Cr(VI) wastewater is usually within the range of 3–5. From a practical standpoint, the optimal pH should be 4 for the proposed synergetic process.

### 2.6. Effects of Co-Existing Ions on Cr(T) Removal

To facilitate the practical application of round-the-clock catalysis for Cr(T) removal, it is indispensable to study the effects of co-existing ions (Cl^−^, SO_4_^2−^, CO_3_^2−^, and Ca^2+^ at a concentration of 5 mM) in aquatic systems, and the results are assessed in Figure 6e,f. As a consequence of the competitive adsorption between Cr species and these inorganic ions, the photoreduction–adsorption efficiency of Cr(VI), as well as the Cr(T) removal, is lower. Therefore, the addition of inorganic ions severely inhibits the photocatalytic rate constant, particularly in the presence of coexisting anions. On one hand, the competitive adsorption between Cr(VI) and Cl^−^, SO_4_^2−^, CO_3_^2−^, etc., on SMSO_ED_/COF_TP-TTA_-10 leads to low photoreduction efficiency under the applied acidic conditions (pH = 4). On the other hand, the pH values change with the addition of Na_2_CO_3_, which directly leads to a decrease in the Cr(VI) reduction rate constant. The apparent rate K of Cr(VI) photoreduction exhibits a decline from 0.067 min^−1^ to 0.021 min^−1^, 0.017 min^−1^, and 0.020 min^−1^ with the addition of Cl^−^, SO_4_^2−^, and CO_3_^2−^, respectively (Appendix A). However, Ca^2+^ ions do not quench the photogenerated carriers due to the high and stable oxidation state, which has no impact on photocatalytic process. However, during the adsorption step, cations (Na^+^ and Ca^2+^) in the solution compete with the resultant Cr(III) ions, weakening the Cr(T) removal capacity of the obtained material.

### 2.7. Possible Photocatalytic Mechanisms

To further investigate the reactive species involved in the Cr(VI) photoreduction process, chemical quenching experiments were performed. In the scavenging test, 1 mM isopropanol (IPA), p-benzoquinone (BQ), potassium bromate (KBrO_3_), and ethylenediaminetetraacetic acid disodium salt (EDTA-2Na) were applied for •OH, O_2_^•−^, e^−^, and h^+^ scavenging, respectively. The reduction rate of Cr(VI) is significantly decreased after the addition of KBrO_3_ (Figure 7a), implying that e- plays a pivotal role in Cr(VI) photoreduction. At the same time, IPA only slightly inhibits Cr(VI) reduction, suggesting that •OH has a mild effect in this reaction process. The addition of EDTA-2Na has been observed to enhance the reduction efficiencies, which can be attributed to the fact that EDTA-2Na acts as a hole-trapping agent in the reduction system. This facilitates the separation acceleration of the photogenerated hole–electron pairs, which ultimately promotes the photoreduction of Cr(VI). Furthermore, the introduction of p-BQ also promotes Cr(VI) photoreduction, implying that the O_2_^•−^ mediated by dissolved O_2_ is not conducive to shifting the equilibrium towards Cr(VI) photoreduction [46]. The above findings can be rationalized by considering the following equations:$$O_{2}+e^{-}\to {O_{2}}^{•-}$$
$${{2O}_{2}}^{•-}+{2H}^{+}\to H_{2}O_{2}+O_{2}$$
$$H_{2}O_{2}+{\mathrm{Cr}}^{3+}+H^{+}\to {\mathrm{Cr}}^{6+}+H_{2}O+•\mathrm{OH}$$

The EPR measurements were carried out to provide direct evidence for the presence of active species. As shown in Figure 7b,c, strong characteristic peaks of superoxide radicals (DMPO—O_2_^•−^) and hydroxyl radicals (DMPO—•OH) are evident after 10 min of light irradiation, substantiating the generation of electrons and holes from SMSO_ED_/COF_TP-TTA_-10. According to the calculated VB of COF_TP-TTA_ and CB of SMSO_ED_, if the SMSO_ED_/COF_TP-TTA_-10 system is suitable for the traditional type-II charge separation mechanism, the holes of COF_TP-TTA_ are unable to oxidize OH^−^ to generate •OH, due to the fact that the VB of COF_TP-TTA_ is more negative than that of •OH (+1.99 eV) [47,48]. The EPR results thus corroborate the construction of direct Z-scheme heterojunction in SMSO_ED_/COF_TP-TTA_-10 composite. Notably, the peaks of these two radicals persist even after the light source is removed for ten minutes. It is owing to the influence of the long afterglow effect of SMSO_ED_ material, which maintains the photocatalytic reactions even after the light irradiation disappears.

Based on the aforementioned findings, a possible mechanism of the round-the-clock photocatalytic Cr(VI) reduction caused by SMSO_ED_/COF_TP-TTA_-10 is proposed, with an illustration diagram provided in Figure 7d. In the constructed composite photocatalytic system, electron transfer follows a Z-scheme electron path, which effectively suppresses the recombination of photogenerated carriers and exhibits excellent photocatalytic activity. As a typical long afterglow material, when Si_2_MgSi_2_O_7_: Eu^2+^, Dy^3+^ is excited under simulated sunlight irradiation, the electrons in the ground state of Eu^2+^ 4f^7^ are activated to the excited state and enter the 4f^6^5d^1^ energy level, thereby generating an autoionization effect and forming free electrons and Eu^3+^. The corresponding free electrons are captured by the Dy^3+^ to form Dy^2+^, storing energy in the process. When the external light source is removed, Dy^2+^ returns to the ground state, releasing electrons which then reduce the Eu^3+^ to Eu^2+^ [49]. The afterglow light, with a wavelength range of 460–550 nm, can be emitted by Si_2_MgSi_2_O_7_: Eu^2+^, Dy^3+^ during this process. The overlap between the range of emission and absorption spectra of SMSO_ED_ and COF_TP-TTA_ allows the afterglow to simulate the SMSO_ED_/COF_TP-TTA_-10 photocatalyst to produce photogenerated electron–hole pairs even when the light is turned off and drive the photocatalytic reactions. The corresponding mechanism is similar to that under simulated sunlight irradiation. A greater number of photogenerated electrons are retained on the CB of COF_TP-TTA_, which facilitates the reduction of Cr(VI). The Cr(III) products are then adsorbed on the surface of the composite through chemical precipitation and electrostatic attraction, thereby achieving total chromium removal.

### 2.8. Active Stability and Reusability Analysis

To examine the stability and regenerability of SMSO_ED_/COF_TP-TTA_-10, three repeated photoreduction cycles were performed at pH = 4 under simulated sunlight irradiation. After each cycle, the sample was separated from the solution, washed with deionized water and then used for the next cycle. As shown in Figure 8a, the catalyst demonstrates remarkable stability and retains high catalytic activity during the whole reaction process. The Cr(VI) reduction rate remains above 92% after three cycles, indicating the catalyst’s consistent efficacy. However, the removal rate of the Cr(T) decreases from 100% to 51.8%, which may be attributed to the occupation of surface-active sites by Cr(III) species. In addition, the results of XRD and SEM show that no significant changes are observed in the crystal structure and morphology before and after the photocatalytic process (Figure 8b). However, the EDS analysis demonstrates the existence of Cr, which is distributed uniformly on the surface of SMSO_ED_/COF_TP-TTA_-10 (Appendix A). Furthermore, the reduction of Cr(VI) and the adsorption of Cr(III) induce some substantial changes in the FT-IR spectra (Figure 8c). A new trough can be observed at 731 cm^−1^ corresponding to Cr-O vibrations resulting from the formation of Cr(III)-O-Cr(VI). This observation aligns with our previous findings [4]. The sharp decrease in the band intensity at 1006 cm^−1^ and 624 cm^−1^ corresponds to Si-O stretching after Cr bonding [50]. The slight changes in the vibration of the triazine ring at 1378 cm^−1^ and 1504 cm^−1^ can be attributed to the involvement of the -NH- group in the adsorption of Cr(III) species. These results indicate that the silica hydroxy and imino groups on the surface of SMSO_ED_/COF_TP-TTA_-10 are instrumental in the chemical binding of Cr species. The results of XPS further reflect the changes in the valence states of the SMSO_ED_/COF_TP-TTA_-10 surface before and after the reaction (Figure 8d). The appearance of Cr element after photoreduction and adsorption reveals that it is undoubtedly involved in the reaction. The peaks at approximately 577.7 and 586.9 eV in the high-resolution spectra of Cr 2p are attributed to Cr(VI), while the peaks at 579.8 and 589.4 eV are ascribed to Cr(III) (Figure 8e). This result also corresponds to the formation of Cr(VI)-O-Cr(III) in FT-IR spectra. Overall, the SMSO_ED_/COF_TP-TTA_-10 catalyst has been proven to be a promising photocatalyst with good reusability.

## 3. Materials and Methods

### 3.1. Materials and Reagents

Pristine vermiculite with a SiO_2_ content of 41.3% was taken from Liaoning, China. MgO (analytical grade), B_2_O_3_ (analytical grade), Dy_2_O_3_ (analytical grade), Eu_2_O_3_ (analytical grade), HCl (analytical grade), and K_2_Cr_2_O_7_ (analytical grade) were obtained from Beijing Chemical Reagent Co., Ltd. (Beijing, China). TP (1,3,5-triformylphloroglucinol, analytical grade) and TTA (4,4′,4″-(1,3,5-triazine-2,4,6-triyl) trianiline, analytical grade) were purchased from Henan Alfa Chemical Co., Ltd. (Zhengzhou, China). DMSO (dimethyl sulfoxide, analytical grade) and NaOH (analytical grade) were obtained from Beijing Chemical Reagent Co., Ltd. All chemical reagents used in this work are obtained from commercial sources and used without further processing.

### 3.2. Preparation of Samples

#### 3.2.1. Preparation of SMSO_ED_ Sample

Sr_2_MgSi_2_O_7_:Eu^2+^, Dy^3+^ powder of 0.3 mol% of Eu^2+^ and 1.5 mol% of Dy^3+^ was prepared by traditional solid-state reaction technique [51]. First, 10 g expanded vermiculite powder was added into hydrochloric acid solution (150 mL, 4 M) and stirred for 12 h to obtain SiO_2_ particles. Then, the obtained SiO_2_ and starting materials Sr(NO_3_)_2_, MgO, B_2_O_3_, Dy_2_O_3_, and Eu_2_O_3_ with proper stoichiometric proportions were thoroughly ground for approximately 30 min in a mortar. The mixture was calcined at 1100 °C for approximately 2 h at a continuous flow of Ar/H_2_ (10 vol%) with a flow rate of 80 mL/min and a heating rate of 5 °C/min.

The relevant chemical reaction is:$$\mathrm{Sr}{\left({\mathrm{NO}}_{3}\right)}_{2}+{\mathrm{SiO}}_{2}+\mathrm{MgO}+{\mathrm{Eu}}_{2}O_{3}+{\mathrm{Dy}}_{2}O_{3}\overset{1100{}^{\circ}C}{\to }{\mathrm{Sr}}_{2}{\mathrm{MgSi}}_{2}O_{7}{:\mathrm{Eu}}^{2+}{,\mathrm{Dy}}^{3+}+{\mathrm{NO}}_{2}↑$$

#### 3.2.2. Preparation of SMSO_ED_/COF_TP-TTA_ Composites

SMSO_ED_/COF_TP-TTA_ composites were prepared by the typical solvent method. A schematic diagram of the synthesis process is provided in Appendix A. Firstly, 5 mg TP and 8.25 mg TTA were added to a beaker containing 10 mL DMSO [52]. After ultrasonication, 200 mg SMSO_ED_ was added into the mixture in a 20 mL Teflon-lined stainless-steel autoclave. After reacting at 160 °C for 24 h, an orange colloidal solution was formed. The obtained solution was washed with DMSO and deionized water and subsequently dried at 60 °C for 12 h to obtain the final product SMSO_ED_/COF_TP-TTA_-10. Similarly, different proportions of TP and TTA were added to obtain SMSO_ED_/COF_TP-TTA_-5 and SMSO_ED_/COF_TP-TTA_-20 (5, 10, and 20 represent the weight percentage of COF_TP-TTA_ in the SMSO_ED_/COF_TP-TTA_ composite materials).

### 3.3. Characterization of Prepared Samples

The morphology of the synthesized samples was observed by scanning electron microscope (SEM, Hitachi (Tokyo, Japan) SU8020) and transmission electron microscope (TEM, JEOL (Tokyo, Japan) JEM-2100). The X-ray diffraction (XRD, Bruker (Billerica, MA, USA) D8) measurements were recorded to analyze the crystal structures. UV-vis diffuse reflectance spectroscopy (DRS, Shimadzu (Tokyo, Japan) UV-3600 plus spectrophotometer) was used to study the optical properties of samples. The surface functional groups were studied by Fourier-transform infrared spectra spectroscopy (FT-IR, PerkinElmer (Waltham, MA, USA) 1730). X-ray photoelectron spectroscopy (XPS, Thermo Fisher (Waltham, MA, USA) ESCALAB 250) was used to analyze the surface chemical composition of the composites. Photoluminescence spectra (PL, LS980 Fluorescence spectrometer, Edinburgh Instruments (Edinburgh, UK)) was used to obtain the emission spectra of the samples with 385 nm excitation wavelength. The electrochemical properties (photocurrent-time and the electrochemical impedance spectroscopy) were performed on an electrochemical workstation (CHI660e, CH Instruments, Shanghai, China). The detection data for free radicals were obtained with electron paramagnetic resonance (EPR, Bruker ESP 500 spectrometer). The concentration of Cr(VI) was determined by the UV-vis absorption spectra, and the concentration of Cr(T) was obtained by inductively coupled plasma-atomic emission spectroscopy (ICP-AES, Optima8300, Schwaebisch Hall, Germany).

### 3.4. Removal Experiments of Cr(T)

The reduction of Cr(VI) and the adsorption of Cr(III) were conducted in a quartz reactor containing 50 mL of Cr(VI) solution with an initial concentration of 10 mg/L, with photocatalyst samples taken at a dosage of 20 mg. The photocatalytic activities of SMSO_ED_/COF_TP-TTA_ were evaluated through the reduction of Cr(VI) under simulated sunlight irradiation. A 300 W Xenon light was used as sunlight source. Before illumination, the suspension was stirred for 30 min under dark conditions to reach adsorption–desorption equilibrium. Subsequently, the whole reaction system was irradiated for 30 min (400 ± 1 mW/cm^−1^), with 2 mL of suspension being collected at 10 min intervals. Then, the light source was turned off and the reaction continued for a further 30 min, with 2.0 mL suspension was taken every 10 min during this process. The reduction rate (%) is calculated according to the following formula:$$C_{t}{/C}_{0}=A_{t}{/A}_{0}$$
where C_0_ is the initial concentration of the Cr(VI) solution, C_t_ is the concentration of the Cr(VI) after reaction; A_0_ is the absorbance of the initial Cr(VI) solution, and A_t_ is the instantaneous absorbance of the Cr(VI) after reaction. The concentration of Cr(III) was obtained by calculating the concentration between Cr(T) and Cr(VI), and the Cr(T) removal efficiency (%) is calculated from:R=C0−CeC0×100%
where C_e_ is the equilibrium concentration of total Cr in the solution.

## 4. Conclusions

In summary, a Si_2_MgSi_2_O_7_: Eu^2+^, Dy^3+^/COF_TP-TTA_-10 composite is synthesized as a novel and efficient self-luminescent photocatalyst for the photoreduction of Cr(VI) and synergetic adsorption of Cr(III) around the clock. The PL spectra and decay curve spectra analyses confirm that the Si_2_MgSi_2_O_7_: Eu^2+^, Dy^3+^/COF_TP-TTA_-10 composite has faster electron transfer at the interface and longer lifetime of charge carriers than single Si_2_MgSi_2_O_7_: Eu^2+^, Dy^3+^ or COF_TP-TTA_. The optimized SMSO_ED_/COF_TP-TTA_-10 sample exhibits an excellent photocatalytic reduction performance of Cr(VI), with a photoreduction efficiency up to 92.8% within 30 min of visible light exposure. The long afterglow effect of SMSO_ED_ enables the composite catalyst to maintain considerable photocatalytic activity for Cr(VI) photoreduction in a sunless condition. After the light source is removed for a period of 30 min, the reduction reaction proceeds and removal efficiency of Cr(T) reaches 97.16%. The resulting radical •OH provides further evidence that the charge separation mechanism is more suitable for a Z-scheme mechanism. The present study is anticipated to provide a strategy for enhancing photocatalytic activity and achieving round-the-clock photocatalysis, which will promote the full industrialization process.

## Figures and Tables

**Figure 1 molecules-29-04327-f001:**
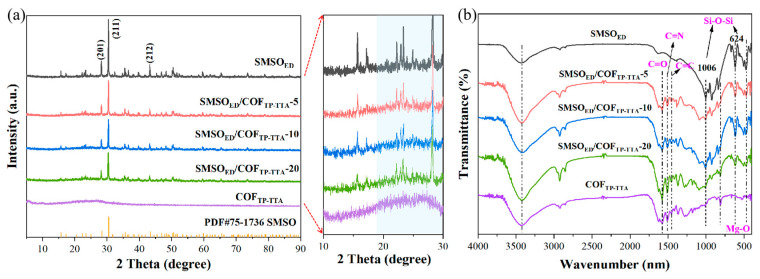
(**a**) XRD patterns and (**b**) FT-IR spectra of SMSO_ED_, COF_TP-TTA_, and SMSO_ED_/COF_TP-TTA_-x (x = 5, 10, and 20) composited photocatalysts.

**Figure 2 molecules-29-04327-f002:**
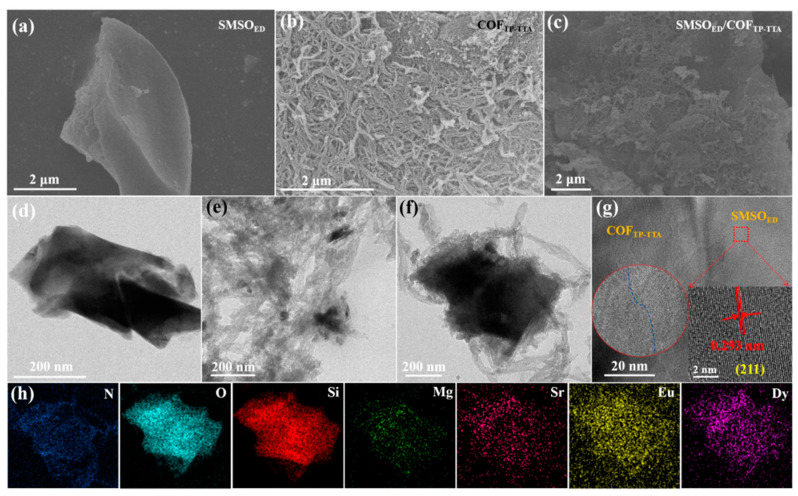
SEM images of (**a**) SMSO_ED_, (**b**) COF_TP-TTA_, (**c**) SMSO_ED_/COF_TP-TTA_-10. TEM images of (**d**) SMSO_ED_, (**e**) COF_TP-TTA_, (**f**) SMSO_ED_/COF_TP-TTA_-10. (**g**) HRTEM image and (**h**) TEM-EDX elemental mapping for SMSO_ED_/COF_TP-TTA_-10.

**Figure 3 molecules-29-04327-f003:**
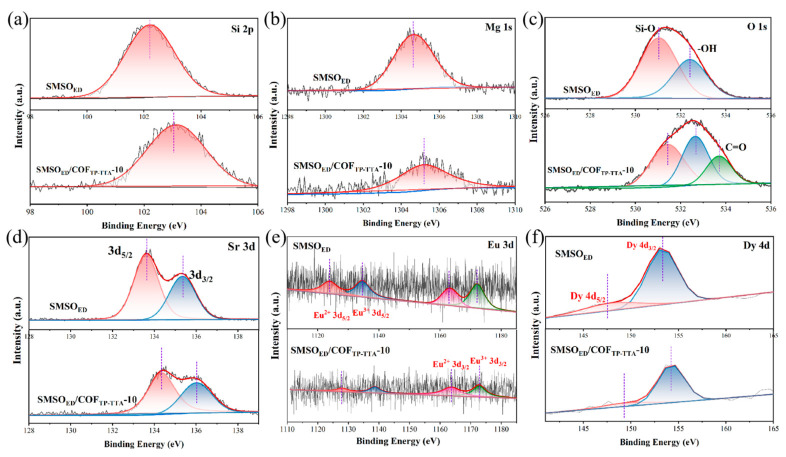
XPS spectra of SMSO_ED_ and SMSO_ED_/COF_TP-TTA_-10. (**a**) Si 2p, (**b**) Mg 1s, (**c**) O 1s, (**d**) Sr 3d, (**e**) Eu 3d, and (**f**) Dy 4d.

**Figure 4 molecules-29-04327-f004:**
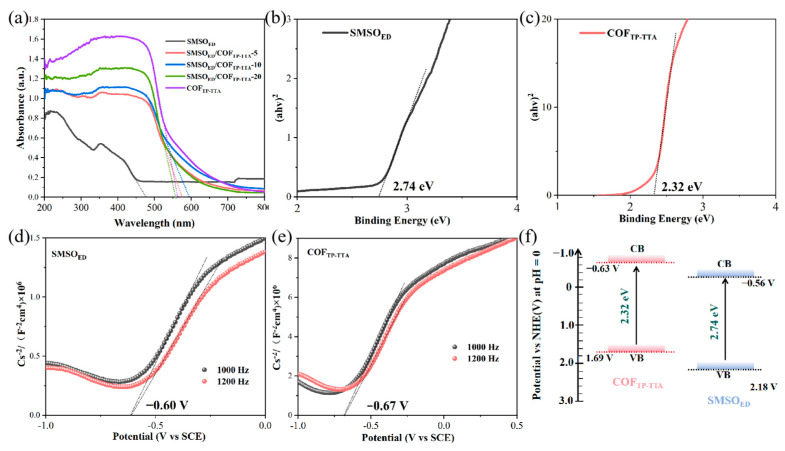
(**a**) UV-vis diffuse reflectance spectra and (**b**,**c**) band gap patterns of SMSO_ED_, COF_TP-TTA_, and SMSO_ED_/COF_TP-TTA_-10. (**d**,**e**) Mott–Schottky plots of SMSO_ED_ and COF_TP-TTA_. (**f**) Schematic diagram of band structure of COF_TP-TTA_ and SMSO_ED_.

**Figure 5 molecules-29-04327-f005:**
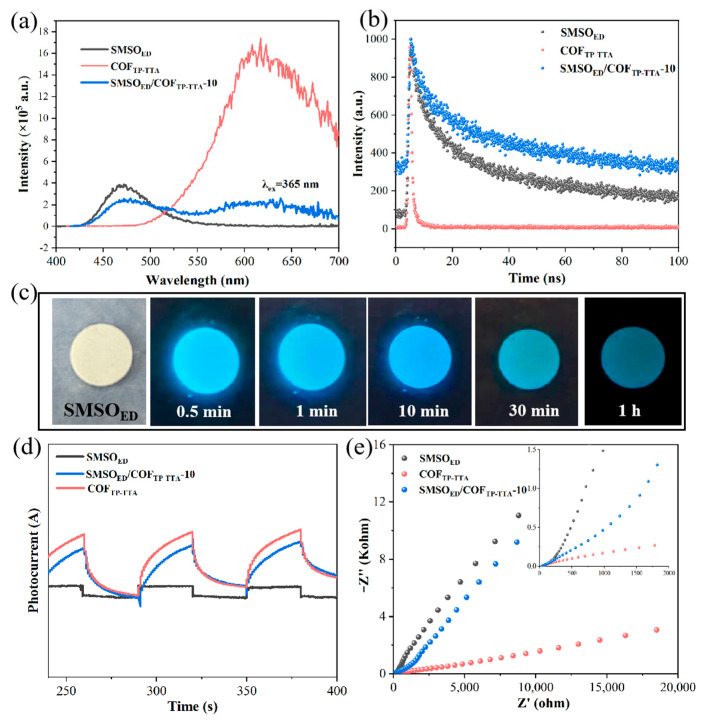
(**a**) PL spectrum and (**b**) decay curves spectra of SMSO_ED_, COF_TP-TTA_ and SMSO_ED_/COF_TP-TTA_-10. (**c**) Real luminescence images of SMSO_ED_ before and after simulated sunlight irradiation. (**d**) Photocurrent–time dependence and (**e**) EIS spectra of SMSO_ED_, COF_TP-TTA_, and SMSO_ED_/COF_TP-TTA_-10.

**Figure 6 molecules-29-04327-f006:**
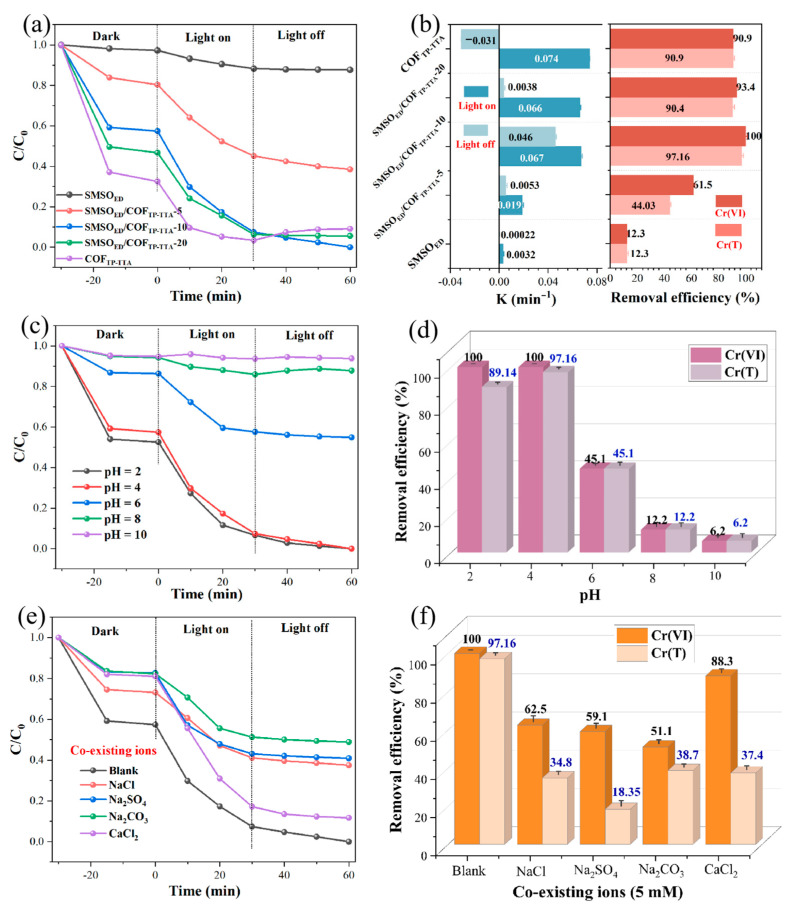
(**a**) Photocatalytic reduction activities towards Cr(VI) with SMSO_ED_, COF_TP-TTA_, and SMSO_ED_/COF_TP-TTA_-x (x = 5, 10, 20) under simulated sunlight irradiation and dark conditions. (**b**) The corresponding kinetic constants and removal rate of Cr(T). (**c**–**e**) Effects of (**c**,**d**) solution pH value and (**e**,**f**) coexisting ions on photoreduction efficiency of SMSO_ED_/COF_TP-TTA_-10. (Conditions: 30 mg SMSO_ED_/COF_TP-TTA_-10, 50 mL 10 mg/L Cr(VI) solution, pH = 4, room temperature.)

**Figure 7 molecules-29-04327-f007:**
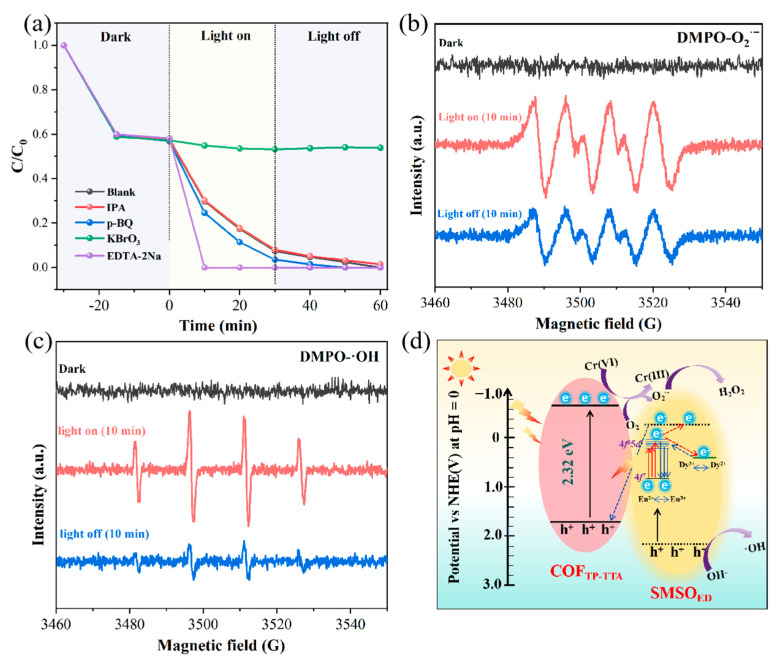
(**a**) Cr(VI) reduction by SMSO_ED_/COF_TP-TTA_-10 in the presence of isopropanol, benzoquinone, KBrO_3_, and EDTA-2Na as radical scavengers. (**b**,**c**) EPR spectra of (**b**) DMPO-O_2_^•−^ and (**c**) •OH under simulated light for 10 min and under dark condition for 10 min. (**d**) The schematic diagram for the possible charge separation and transfer mechanism of SMSO_ED_/COF_TP-TTA_-10.

**Figure 8 molecules-29-04327-f008:**
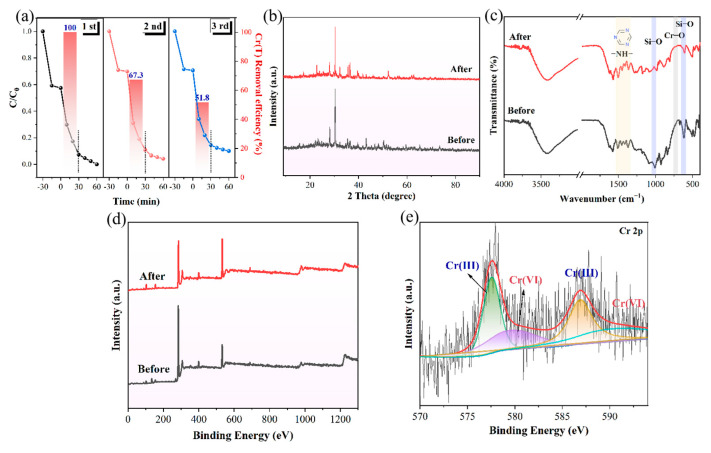
(**a**) Cycling experiments of photoreduction Cr(VI) over SMSO_ED_/COF_TP-TTA_-10. (**b**) XRD, (**c**) FT-IR, and (**d**) XPS survey patterns of SMSO_ED_/COF_TP-TTA_-10 before and after photocatalysis. (**e**) Cr 2p high-resolution spectra of SMSO_ED_/COF_TP-TTA_-10 after photocatalysis.

## Data Availability

Data are contained within the article and Appendix A.

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
