# Peer review of "Fabrication of Active Z-Scheme Sr2MgSi2O7: Eu2+, Dy3+/COF Photocatalyst for Round-the-Clock Efficient Removal of Total Cr"

_molecules, 2024, doi:10.3390/molecules29184327_

Round 1

Reviewer 1 Report

Comments and Suggestions for Authors

This work developed a Z-scheme Si2MgSi2O7: Eu2+, Dy3+/COFTP-TTA heterostructure for the photoreduction of Cr(VI) and synergetic adsorption of Cr(III) around the clock. The investigation on long-afterglow catalysis is interesting. Therefore, I recommend its publication after addressing the following issues.

1. In the Abstract, the authors said “functional groups (amino, or hydroxyl) on the composited catalyst can adsorb the resultant Cr(III) species”, where are these functional groups? The authors should indicate these groups by FTIR and XPS.

2. Fig. 2b/c/e/f should be marked when describing them in the manuscript.

3. Lots of writing mistakes must be corrected, e.g. lines 188, 189, 190, 241, 260, 519; the captions of Fig. 7 and 8 are the same; the Cr(T) removal efficiency in the first time should be 97.16%.

4. The corresponding groups should be marked in Fig. 8c.

5. Zeta potentials of SMSOED/COFTP-TTA-10 composites should be tested and discussed rather than COFTP-TTA.

6. Real luminescence images of SMSOED/COFTP-TTA-10 composites before and after simulated sunlight irradiation should be also provided.

7. The volume of the reaction solution should be given.

8. The part “Conclusions” should be revised, the description of “The binding energy shifts of the Si 2p peaks confirm the successful construction of an intimate interface” is not suitable.

9. Some relative studies on COF-based photocatalysts (AIChE J. 69 (2023) e18192; Colloid Surf. A 664 (2023) 131124; Catal. Sci. Technol. 14 (2024) 590-597) could be read to improve the quality of this manuscript.

Comments on the Quality of English Language

Minor editing of English language required.

Reviewer 2 Report

Comments and Suggestions for Authors

This study introduces a novel approach for developing all-weather Cr(VI) photocatalysts, effectively addressing the challenges posed by intermittent solar flux and limited light absorption. By incorporating Sr2MgSi2O7:Eu2+ and Dy3+ as active sites and electron transfer centers within COF molecules, significant potential has been demonstrated. Additionally, the combined influence of the interlayer structure and functional groups on the composite catalyst plays a crucial role in enhancing the comprehensive removal of Cr, thereby increasing the practicality and significance of this research. Therefore, I recommend its publication in the journal "Molecules" after addressing the following minor issues.

1.      In the manuscript, figures 2a and 2b mentioned in lines 144 and 145 respectively exhibit the SEM and TEM images of the SMSOED composite material. Can this explain the morphology of the SMSOED/COFTP-TTA composite material? Please provide clearer photos to illustrate the composition of the composite material.

2.      In the manuscript, the peak at 533.9 eV in line 170 should be attributed to the C=O bond, while in Figure 3, the O 1s peak should be labeled as the C-O bond. Please specify the corresponding figure number to ensure consistency between the text and the figures.

3.      In the 200 and 201s lines, 0.36 eV and 0.43 eV are positive, while in Figures 4d and 4e, they are negative. Please carefully check if the subsequent sets of data are correct and make corrections if there are errors.

4.      In lines 368-372, it should be demonstrated as a Z-scheme heterojunction instead of the type-II charge separation mechanism. Please cite references for further verification.

5.      In lines 138 and 139, the absence of a newly introduced functional group implies that the interaction between COFTP-TTA and SMSOED within the composite material occurs at the interface rather than through chemical bonding. The specific functional group that is missing to indicate this interface interaction needs to be identified. Please provide references to support this assertion would strengthen the explanation.

6.      Figures 5d and 4f in lines 320 and 340 do not match the description in the manuscript, please carefully review and correct.

Reviewer 3 Report

Comments and Suggestions for Authors

Round 2

Reviewer 3 Report

Comments and Suggestions for Authors

The authors have addressed my question adequately.